# Immunization and SARS-CoV-2 Antibody Seroprevalence in a Country with High Vaccination Coverage: Lessons from Chile

**DOI:** 10.3390/vaccines10071002

**Published:** 2022-06-23

**Authors:** Ximena Aguilera, Claudia González, Mauricio Apablaza, Paola Rubilar, Gloria Icaza, Muriel Ramírez-Santana, Claudia Pérez, Lina Jimena Cortes, Loreto Núñez-Franz, Rubén Quezada-Gaete, Carla Castillo-Laborde, Juan Correa, Macarena Said, Juan Hormazábal, Cecilia Vial, Pablo Vial

**Affiliations:** 1Centro de Epidemiología y Políticas de Salud, Facultad de Medicina Clínica Alemana, Universidad del Desarrollo, San Carlos de Apoquindo, Las Condes, Santiago 7610658, Chile; claudiagonzalez@udd.cl (C.G.); paolarubilar@udd.cl (P.R.); carlacastillo@udd.cl (C.C.-L.); 2Facultad de Gobierno, Universidad del Desarrollo, San Carlos de Apoquindo, Las Condes, Santiago 7610658, Chile; mapablaza@udd.cl; 3Instituto de Matemáticas, Universidad de Talca, Talca 3460000, Chile; gicaza@utalca.cl; 4Public Health Department, Faculty of Medicine, Universidad Católica del Norte, Coquimbo 1780000, Chile; mramirezs@ucn.cl (M.R.-S.); rquezada@ucn.cl (R.Q.-G.); 5Escuela de Enfermería, Facultad de Medicina Clínica Alemana, Universidad del Desarrollo, Santiago 7610658, Chile; claudiaperez@udd.cl; 6Instituto de Ciencias e Innovación en Medicina, Facultad de Medicina Clínica Alemana, Universidad del Desarrollo, Santiago 7610658, Chile; linacortes@udd.cl (L.J.C.); jhormazabal@udd.cl (J.H.); mcvial@udd.cl (C.V.); pvial@udd.cl (P.V.); 7Departamento de Salud Pública, Facultad de Ciencias de la Salud, Universidad de Talca, Avenida Uno Poniente #1141, Talca 3460000, Chile; lnunezf@utalca.cl (L.N.-F.); macarena.said@utalca.cl (M.S.); 8Centro Producción del Espacio, Universidad de las Américas, Santiago 7500975, Chile; juan.correaparra1@gmail.com

**Keywords:** COVID-19, vaccines, Sinovac Coronavac, BNT162b2, AZD1222, ELISA, cross-sectional

## Abstract

Chile is among the most successful nations worldwide in terms of its COVID-19 vaccine rollout. By 31 December 2021, 84.1% of the population was fully vaccinated, and 56.1% received booster doses using different COVID-19 vaccines. In this context, we aimed to estimate the prevalence of anti-SARS-CoV-2 antibodies following the infection and vaccination campaign. Using a three-stage stratified sampling, we performed a population-based cross-sectional serosurvey based on a representative sample of three Chilean cities. Selected participants were blood-sampled on-site and answered a short COVID-19 and vaccination history questionnaire using Wantai SARS-CoV-2 Ab ELISA to determine seroprevalence. We recruited 2198 individuals aged 7–93 between 5 October and 25 November 2021; 2132 individuals received COVID-19 vaccinations (97%), 67 (3.1%) received one dose, 2065 (93.9%) received two doses, and 936 received the booster jab (42.6%). Antibody seroprevalence reached 97.3%, ranging from 40.9% among those not vaccinated to 99.8% in those with booster doses (OR = 674.6, 154.8–2938.5). SARS-CoV-2 antibodies were associated with vaccination, previous COVID-19 diagnosis, age group, and city of residence. In contrast, we found no significant differences in the type of vaccine used, education, nationality, or type of health insurance. We found a seroprevalence close to 100%, primarily due to the successful vaccination program, which strongly emphasizes universal access.

## 1. Introduction

Latin American countries have been severely impacted by the COVID-19 pandemic, partly due to health and socioeconomic inequities and political instability [1], and Chile was not an exception. By the end of June 2020, the country had one of the worst outbreaks in the world. However, the prompt implementation of an aggressive vaccination strategy [2,3] and population compliance [4] transformed the country into a successful example of the COVID-19 vaccine rollout.

The vaccination process started in late December 2020 with health workers and teachers. In February, the massive vaccine rollout began prioritizing the elderly population [5]. By the end of 2021, 84.1% of the Chilean population had been fully vaccinated, and 56.1% had received the booster dose. Despite the rapid advance in vaccination during the second great wave of COVID-19, the outbreak of SARS-CoV-2 associated with the Lambda and mainly Gamma variants could not be controlled [6]. However, the later circulation of the Delta variant, predominantly during the last southern spring (October–December), did not reach the magnitude of previous waves.

One of the striking aspects of the vaccination campaign in Chile is the concomitant use of different vaccine technologies. Of 44.4 million doses administered in 2021 (up to 4 January 2022), 55.8% were the inactivated Sinovac’s CoronaVac. Nucleic Acid mRNA vaccines represented 35.8% (Pfizer-BioNTech’s BNT162b2), and 8.4% was viral vector vaccines (7.1% Oxford-AstraZeneca’s AZD1222, and 1.3% CanSino Biologics’ Ad5-nCoV).

The most frequent vaccine schedule used in Chile at the moment of the study was heterologous and differed by age group. In people aged 55 and older, combined two doses of inactivated vaccine with a booster viral vector vaccine; in youngers, the booster was with the mRNA vaccine. The second most frequent scheme was homologous, using two doses and booster of mRNA vaccines (BNT162b2). Since September 2021, the health authority extended the vaccination campaign to children between 6 and 17 and from December to children from 3 years of age with CoronaVac for both groups. Two weeks later, the health authority approved BNT162b2 for children aged five and older.

This study aims to estimate the seroprevalence of antibodies in individuals with SARS-CoV-2 due to natural or vaccine-induced infection. Our main contribution to the literature is our use of a population-based serosurvey in an emerging country with high vaccine coverage and various vaccine schemes.

## 2. Materials and Methods

We performed a population-based cross-sectional serosurvey based on a previous study of a representative sample of the cities of Santiago, Coquimbo/La Serena, and Talca [7]. This study used a stratified sampling of three stages, census district, block, and dwelling for Coquimbo–La Serena and Talca, and municipality, block, and dwelling for the city of Santiago. All former participants were contacted, and those who refused to participate or were not located were replaced in a standardized manner. Each participant answered a questionnaire, and a blood sample was taken through venipuncture to determine antibodies through the Wantai SARS-CoV-2 Ab ELISA test following the manufacturer’s protocol. For those who presented difficulties with venipuncture, we offered a lateral flow immunoassay, which required capillary blood sampling (Cellex qSARS-CoV-2 IgG/IgM Cassette Rapid Test kit; compared to Rt-PCR-positive, percent agreement is 93.8%, and negative percent agreement is 96%).

Participants completed an on-site questionnaire regarding basic demographic and socioeconomic variables, self-reported COVID-19 infection by asking for PCR confirmation, history of COVID-19 exposure, symptoms, and questions about vaccination (the number of doses, type of vaccine, and administration time). Seroprevalence was expressed as a proportion of the sampled population. The association between seroprevalence and risk factors or vaccine use was expressed as odds ratios (ORs) with a 95% confidence interval. Data were analyzed using STATA statistical software (StataCorp. 2017. Stata Statistical Software: Release 15. StataCorp LLC, College Station, TX, USA). This research followed the generic protocol of World Health Organization Unity studies [8,9]. Protocols were approved by the Ethics Committee of Universidad del Desarrollo, Universidad Catolica de Norte, and Universidad de Talca.

## 3. Results

Between October 5th and November 25th, 2021, 2198 individuals in 1131 dwellings were recruited in Santiago (1204–54.8%), Coquimbo–La Serena (536–24.4%), and Talca (458–20.8%). A quarter of the households in the first study were not recruited due to refusal or a change of residence. These cases were replaced by recruiting new dwellings, following the same sampling method. The average age of the participants was 43.7 years (SD 19.1), and 833 were males (37.9%). We collected 2176 serum samples and used rapid tests on 22 participants (Table 1).

The overall sample seroprevalence was 97.3%, which was significantly higher in Santiago and Talca than in Coquimbo–La Serena (98.4%, 98.5%, and 93.9%, respectively). The odds ratio (OR) provides information to compare the seroprevalence of each category of risk factors, including vaccines, using one of them as a reference. As seen in Table 1, there were no significant differences between sexes, and concerning age, individuals aged less than 10 years had a significantly lower seroprevalence than other age groups. From the age of 20, the seroprevalence exceeded 97%, reaching almost 100% in the 30–39 group. Previous COVID-19 history (self-reported) was present in 12.1% of the sampled individuals. Among them, the presence of antibodies was approximately 100% and was significantly higher than in people who did not have this antecedent. Previous COVID-19 history was more frequent in people between 30 and 60 years old than in those younger and older.

Concerning vaccine coverage, 97% of the enrolled subjects had received at least one dose of the COVID-19 vaccine, and this figure was higher in Santiago and Talca than in Coquimbo–La Serena and in people older than 20 years old. There was a significant association between antibodies and vaccination history (Table 2). Seroprevalence doubled in people with one dose compared to those not vaccinated and increased with more doses, reaching almost 100% in subjects who received the booster jab (Mantel–Haenszel chi-square = 395.5030; *p*-value < 0.0001). Having two doses (or a complete basal schedule using Ad5-nCoV) increases the probability of having antibodies by 12 times compared with one dose while having a booster dose increases it by 52 times.

The seroprevalence differences in the time elapsed between the last dose of vaccine and the blood sample obtained for this study were not significant. However, the highest seroprevalence was between 15 and 180 days (Table 3).

## 4. Discussion

The results show a high seroprevalence of antibodies in the subjects studied, strongly associated with vaccination, consistent with the immunization coverage registered by the Chilean Ministry of Health for the three cities. At the end of October, during the fieldwork, reported vaccine coverage was 83.8% for the first dose, 76.9% for two doses, and 29.9% for the booster jab [10]. The cumulative incidence reported by the Chilean MOH at that time was 7059 per 100,000 inhabitants in Coquimbo–La Serena; 11,724 in Santiago; and 10,076 in Talca. The SARS-CoV-2 Delta variant represented 97% of the cases. Figure 1 shows the COVID-19 epi-curve in the studied cities, indicating the fieldwork period; Figure 2 shows the results of genomic surveillance of SARS-CoV-2 variants of concern performed by the Institute of Public Health in Chile [6].

Our results show a higher seroprevalence than that reported in other population-based seroprevalence studies. In India, between June and July 2021, seropositivity in adults was 66.7% [11]; in Switzerland, in the same period, it reached 64.4%, with 30% due to natural infection [12], while in Greece in June 2021, it was 55.7% [13]. This difference may be due to the high vaccination coverage achieved in Chile. On the other hand, the study carried out in India, similar to our results, shows an antibody gradient according to the number of vaccine doses (62% among those not vaccinated; 81% with one vaccine dose, and 89.8% with a full scheme) [11].

Coquimbo–La Serena showed a significantly lower seroprevalence than the other two cities, which could be explained by the higher proportion of unvaccinated individuals in that city obtained in our study (5%, versus 2.5% in Santiago and 2% in Talca). Another factor could be that the accumulated incidence in Coquimbo was lower than that in the other cities at the time of the study.

The lower seroprevalence observed in children under 10 years of age is probably due to their late incorporation into the vaccination campaign, which as of 27 September 2021, covered children between 6 and 11 years of age. We also found an association between seroprevalence and history of COVID-19, but with lower significance than vaccination.

The lack of association of seroprevalence with education, health insurance (public or private), and nationality is consistent with the strategy of the health authority. The early purchase of jabs and the implementation of a universal vaccination campaign allowed access to all people, taking care of vulnerable and hard-to-reach populations, including migrants. In contrast, the Geneva study showed significant differences in vaccination coverage related to education, leading to higher seroprevalence among more educated people [12].

The seroprevalence among vaccinated subjects (any dose) reached 99.1%, while that among unvaccinated subjects was 40.9%. We observed a gradient effect between the prevalence of antibodies and the number of vaccine doses received. Those with the booster doses reach almost 100% seroprevalence, independent of the type of vaccine used.

Regarding the type of vaccines applied, the most significant antibody variation occurred in those who received only one dose. In contrast, those who completed the two-dose schedules had a similar seroprevalence, regardless of the type of vaccine administered or the age-related vaccine schedule defined by the health authority. The mRNA-exclusive scheme was available to health care workers in intensive care units at the beginning of the vaccine rollout. Later there was open to the general public when most elders were already vaccinated. Then, the people who received mRNA are on average younger than those who received the heterologous regimen. Finally, seroprevalence was even more homogeneously high in those with booster doses.

The prevalence of antibodies in unvaccinated people demonstrates the burden of infection achieved with successive pandemic waves. In our previous survey, at the end of 2020, the seroprevalence was 10.4%, with differences between the three cities. There are still differences among the cities, although they all have very high seroprevalence.

Our study had some limitations. The rejection rate of former participants could be associated with the vaccination status; therefore, our results might have shown higher vaccine uptake than officially reported. However, this does not preclude comparative analysis between vaccinated and unvaccinated participants and the different vaccine schemes. Further analysis will include weighted seroprevalence and neutralizing antibody responses.

In sum, our results show a prevalence of antibodies close to 100% in the population aged seven years and older, primarily due to the successful vaccination program, which strongly emphasizes universal access. We also observed that seroprevalence remained very high 180 days after the last vaccine dose. However, this might not be enough to prevent the recent summer resurgence of cases resulting from the circulation of the Omicron variant, even though it is expected to have less impact on hospitalizations and deaths. We still need to learn about SARS-CoV-2 transmission, the protection conferred by vaccines, and their relationship with the evolution of the virus.

## Figures and Tables

**Figure 1 vaccines-10-01002-f001:**
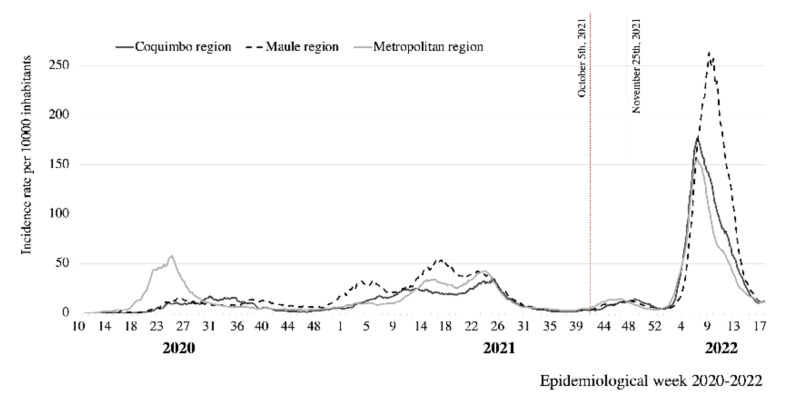
COVID-19 epidemic curve in three regions of Chile, 2020–2022. Source: [10].

**Figure 2 vaccines-10-01002-f002:**
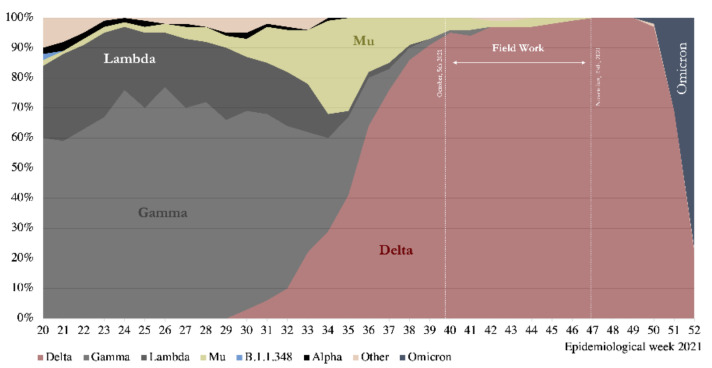
Genomic Surveillance for SARS-CoV-2 Variants of Concern circulating in Chile, 2021. Source: [6].

**Table 1 vaccines-10-01002-t001:** Total sample SARS-CoV-2 seroprevalence by subgroup.

		Number of Observations	Seropositive Cases	Percentage	OR (CI 95%)
Total		2198	2139	97.30%	
City	Coquimbo–La Serena	536	503	93.80%	Ref.
	Talca	458	451	98.50%	4.2 (1.9–9.6)
	Santiago	1204	1185	98.40%	4.1 (2.3–7.3)
Sex	Males	833	805	96.60%	Ref.
	Females	1365	1334	97.70%	1.5 (0.9–2.5)
Age	Less than 10 years	32	22	68.80%	Ref.
	10–19 years	242	225	93.00%	6.0 (2.5–14.7)
	20–29 years	321	315	98.10%	23.9 (7.9–71.7)
	30–39 years	358	357	99.70%	162.3 (19.9–1325.4)
	40–49 years	370	363	98.10%	23.6 (8.2–67.9)
	50–59 years	367	359	97.80%	20.4 (7.3–56.8)
	60–69 years	309	303	98.10%	23.0 (7.6–69.0)
	70 or more	199	195	98.00%	22.2 (6.4–76.6)
Nationality	Chilean	2118	2061	97.30%	Ref.
	Foreign	80	78	97.50%	1.1 (0.3–4.5)
Health Insurance	Public (FONASA)	1639	1588	96.90%	Ref.
(*n* = 2141)	Private (Isapre)	455	447	98.20%	1.8 (0.8–3.8)
	Armed Forces	47	47	100.00%	-
Self-reported	No	1931	1873	97.00%	Ref.
COVID-19 diagnosis	Yes	267	266	99.60%	8.2 (1.1–59.7)
Education	Primary	181	176	97.20%	Ref.
18 years and more	High School	1011	990	97.90%	1.3 (0.5–3.6)
(*n* = 1966)	Technical education	275	272	98.90%	2.6 (0.6–10.9)
	Professional	499	494	99%	2.8 (0.8–9.8)

**Table 2 vaccines-10-01002-t002:** SARS-CoV-2 seroprevalence by number of doses and type of vaccine used.

		Number of Observations	Seropositive Cases	Percentage	OR (CI 95%)
Unvaccinated population		66	27	40.90%	Ref.
At least one doses		2132	2112	99.10%	152.5 (78.9–294.9)
One dose:		67	60	89.60%	12.4 (5.0–31.2)
	CoronaVac	58	52	89.70%	
	BNT162b2	6	5	83.30%	
	Other vaccine	3	3	100%	
Two doses or complete basal schedule:		1129	1118	99.00%	146.8 (67.9–317.2)
	CoronaVac	600	591	98.50%	
	BNT162b2	442	440	99.60%	
	Other combination or Ad5-nCoV	87	87	100%	
Complete basal plus booster		936	934	99.80%	674.6 (154.8–2938.5)
	CoronaVac (2) + AZD1222 booster	358	357	99.70%	
	CoronaVac (2) + BNT162b2 booster	359	359	100.00%	
	CoronaVac (2) + CoronaVac booster	103	102	99.00%	
	BNT162b2 (2) + BNT162b2 booster	102	102	100%	
	Other combination	14	14	100%	

**Table 3 vaccines-10-01002-t003:** Seroprevalence of SARS-CoV-2 by time elapsed since the last dose of vaccine.

Days since the Last Dose of the Vaccine	Number of Observations	Seropositive Cases	Percentage (CI 95%)
Less than 15 days	142	139	97.9 (94.0–99.6)
15–180 days	1682	1670	99.3 (98.8–99.6)
More than 180 days	308	303	98.4 (96.3–99.5)

## Data Availability

Study datasets are available in a data repository (https://doi.org/10.5281/zenodo.5773152) (accessed on 13 March 2022) and/or available from the authors upon reasonable request.

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
