# Peer review of "Immunization and SARS-CoV-2 Antibody Seroprevalence in a Country with High Vaccination Coverage: Lessons from Chile"

_vaccines, 2022, doi:10.3390/vaccines10071002_

Round 1
Reviewer 1 Report
The authors have prepared a careful study and analysis of the impact of vaccination on seroprevalence in the Chilean population as sampled in 3 distinct areas. I would recommend the addition of a figure that contains a timeline for when such studies were taken and compares it with the prevalence of COVID-19 strains and also incidence in the regions studied. This could provide a good background comparison for the status of the virus in those regions at the time of the studies reported here and could be significant to understand not just the seroprevalence but its actual impact on COVID 19 strains. The authors have carefully documented the specifics of individual vaccinations, including boosters, across 3 specific regions and included analysis of demographics, e.g; age, etc. In addition they have included analysis of those individuals who had COVID 19 infection although they do not describe whether this is self-reported and/or verified with actual testing. There are several additional comparisons that would appropriate to help further analysis of this complex disease presentation. 1) what are the rates of infection within each of the 3 regions, 2) over the period of the study, which and 3) when were variants prevalent. 4) was testing done for specific VOCID variants?, 5) for the vaccines being studied, which variants were they developed to address? 6) were these all mRNA vaccines? 7 was there a difference between patients receiving different vaccines or combinations of vaccines? |
Author Response
Response to Reviewer 1 Comments
The authors have prepared a careful study and analysis of the impact of vaccination on seroprevalence in the Chilean population as sampled in 3 distinct areas.
Point 1: I would recommend the addition of a figure that contains a timeline for when such studies were taken and compares it with the prevalence of COVID-19 strains and also incidence in the regions studied. This could provide a good background comparison for the status of the virus in those regions at the time of the studies reported here and could be significant to understand not just the seroprevalence but its actual impact on COVID 19 strains.
Response 1: Thanks for the recommendation. We added a figure (Figure 1) showing the COVID-19 epi curve in the studied cities, indicating the fieldwork period (line 148).
Point 2: The authors have carefully documented the specifics of individual vaccinations, including boosters, across 3 specific regions and included analysis of demographics, e.g; age, etc. In addition they have included analysis of those individuals who had COVID 19 infection although they do not describe whether this is self-reported and/or verified with actual testing.
Response 2: Thanks for the observation; we realize that we should have described COVID-19 status as self-reported by asking for PCR confirmation for more clarity. We specified that the COVID-19 Infection status was “self-reported”, in the Materials and Methods and Results section, as well as in Table 1 (line 88, 113, and first column table 1). Nevertheless, the self-report was based on test results at diagnosis.
Point 3: There are several additional comparisons that would appropriate to help further analysis of this complex disease presentation. 1) what are the rates of infection within each of the 3 regions, 2) over the period of the study, which and 3) when were variants prevalent. 4) was testing done for specific VOCID variants?, 5) for the vaccines being studied, which variants were they developed to address? 6) were these all mRNA vaccines? 7 was there a difference between patients receiving different vaccines or combinations of vaccines?
Response 3: Thanks for the suggestions. Concerning 1, we added each regional cumulative incidence notified to the Chilean MOH in the first paragraph of the Discussion section (lines 141-143). 2) Also, we added a figure 2, showing the results of genomic surveillance of SARS-CoV-2 variants of concern between epidemiological weeks 20-52 of 2021, performed by the Institute of Public Health in Chile(line 145). 3) Following your suggestion, we specified that our study period coincided with the predominance of the Delta variant (line 143). 4) We did not perform any testing for VOC variants in this study, we mentioned the lack of neutralizing antibodies testing as a limitation of this study (line 208). However, the Chilean Institute of Public Health, which performs those analyses for travelers and community cases, reported that 97% of the sequenced cases at the moment of the study were Delta VOC (available at https://vigilancia.ispch.gob.cl/app/varcovid). 5) All the vaccines studied and applied in Chile were developed to address the ancestral type of SARS-CoV-2. 6) In the introduction section we rewrite the sentences where we explained the vaccines in use, mentioning with more clarity that at the moment of our study we had three different types of vaccines, inactivated, mRNA, and Viral-vector vaccines (lines 55-59). Also, we added that the most common regime at the moment of the study was heterologous (lines 61), and in the discussion section we give more detail on the age-related vaccine schedule used in Chile (lines 194-200).
Reviewer 2 Report
The high response to seroconversion after COVID vaccination in Chile provides good regional data that could be useful in global analyses of the response to COVID especially alongside the data from other countries. This well discussed. There are however a few points that were hard to understand
Line 77. It is not clear how the "Cellex 77 qSARS-CoV-2 IgG/IgM Cassette Rapid Test kit" helps people adverse to venepuncture because blood is needed . It seems the paper might mean adverse to the nasal sampling and RT-PCR or another test.
Line 93-94. "A quarter of the households in the origininal.........." The meaning of this sentence is unclear. Perhaps it means that a quarter of the samples were not obtained.
The paper does not explain the odds ratios. Ratio of what?
40.9% of the unvaccinated were seropositive whereas the figure given for COVID infection the population was 12%. Does this mean many of the subjects that had COVID did not present for vaccination?
Author Response
Response to Reviewer 2 Comments
The high response to seroconversion after COVID vaccination in Chile provides good regional data that could be useful in global analyses of the response to COVID especially alongside the data from other countries. This well discussed. There are however a few points that were hard to understand
Point 1: Line 77. It is not clear how the "Cellex 77 qSARS-CoV-2 IgG/IgM Cassette Rapid Test kit" helps people adverse to venepuncture because blood is needed . It seems the paper might mean adverse to the nasal sampling and RT-PCR or another test.
Response 1: The determination of antibodies is carried out in blood, taken through venipuncture. In cases where there were problems taking the sample, a rapid test was used with the collection of a drop of capillary blood. Changes were made to the first paragraph of the methods section to better explain (line 84)
Point 2: Line 93-94. "A quarter of the households in the original.........." The meaning of this sentence is unclear. Perhaps it means that a quarter of the samples were not obtained.
Response 2: The comment is appreciated. The phrase tries to explain that this study presents the results of a second evaluation that the research team carries out in the same cities. The explanation was rephrased for more clarity (lines 101-103)
Point 3: The paper does not explain the odds ratios. Ratio of what?
Response 3: We thank the comment. We added a sentence in the Results section saying “The Odds ratio (OR) provides information to compare the seroprevalence of each category of risk factors, including vaccines, using one of them as a reference” (line 108-109).
Point 4: 40.9% of the unvaccinated were seropositive whereas the figure given for COVID infection the population was 12%. Does this mean many of the subjects that had COVID did not present for vaccination?
Response 4: Thanks, we realize we did not specify that the COVID-19 status was self-reported (confirmed diagnosis with PCR). Twelve percent was the proportion of the studied subjects who self-reported COVID-19. For more clarity, we specified this in the materials and methods (line 88), Results (line 113), and table 1. Usually, self-reported cases are lesser than the real infected ones due partly to the presence of asymptomatic or oligosymptomatic infections and access barriers to testing. In our previous seroprevalence studies, the ratio between infected to detected cases was 3:1. Additionally, the symptomatic disease is not related to the vaccine status. In our study, 97% of the subjects received at least one vaccine; this figure was 96.6% among those who self-reported COVID-19.